# Belief in science-related conspiracy theories is not just a matter of knowledge: The democratic quality of countries as a protective factor

**Irene López-Navarro** [ORCID]◉*, **Libia Santos-Requejo** [ORCID]◉*

Universidad de Salamanca, Instituto de Estudios de la Ciencia y la Tecnología, Edificio I+D+i C/Espejo, Salamanca, Spain

◉ These authors contributed equally to this work.
* irene.lopez@usal.es (ILN); libia@usal.es (LSR)

## Abstract

Concerns regarding the increasing prevalence of conspiracy theory beliefs are growing in public discourse. This study investigates science-related conspiracy theories due to their significant potential impact on individual and global health. Using data from a Special Eurobarometer survey (n = 37,079), our research demonstrates the importance of considering public perceptions of science when trying to understand the acceptance of such conspiracy theory beliefs. They are strongly associated with mistrust in scientists and lower level of knowledge. But there is a key factor at the country level, the quality of democracy, which plays a protective role against the acceptance of unwarranted beliefs. Even individuals with lower levels of knowledge or greater mistrust of scientists are less likely to believe in science-related conspiracy theories if they live in democratic countries than in authoritarian states.

## Introduction

### Background and objectives

The rise of anti-scientific attitudes related to issues such as health, demography or climate change have been present in Western societies for years. However, in a post-pandemic context characterised by a worrying increase in misinformation [1] and a rise of populist strategies in the political arena [2], concern about the increasing percentage of people who believe in conspiracy theories is becoming prominent in the public agenda. Recent studies have warned of the increasing prevalence of conspiracy theory beliefs in different countries [3] and the relative stability of their support [4]. According to the latest Eurobarometer on attitudes to science, 28% of Europeans agreed with the idea that viruses have been produced in government laboratories to control our freedom [5]. In the United States more than 35% of survey respondents agreed with the statement that FDA is deliberately preventing the public from getting natural cures for cancer due to commercial interests [6]. Given

**Data availability statement:** All data files are available from GESIS repository (accession number https://doi.org/10.4232/1.13884).

**Funding:** This paper is part of the project 'The role of distributed and dialogic expertise in the resolution of public scientific-technological controversies: an epistemological, argumentative and sociological analysis'. It has been funded by the Spanish Ministry of Science and Innovation. Code PID2019-105783 GB-I00.

**Competing interests:** The authors have declared that no competing interests exist.

that the single best predictor of belief in one conspiracy theory is belief in a different conspiracy theory [7], it is not surprising that some authors have pointed to a general propensity towards conspirational thinking behind these data as a generalised political attitude [8].

Conspiracy theory is a "proposed explanation of some historical event (or events) in terms of the significant causal agency of a relatively small group of persons—the conspirators—acting in secret" [9, p.116]. This elite of powerful actors could be governments, ethnic groups, religious groups, senior businessmen or scientists. All of them have in common supposed resources (economic, knowledge, political influence, information) that give them a great capacity to act (always in a hidden way) in favour of their own benefit and against people outside this elite (i.e., the ordinary people). In addition to this anti-elitist component, conspiracy theories are also inherently anti-systemic, as they must conflict with official or institutional explanations [10,11].

Science-related conspiracy theories (SCT) constitute a specific subtype of these unfounded beliefs. While sharing the same anti-elitist and anti-establishment character, they are particularly relevant because they question traditional forms of epistemic authority rooted in scientific institutions. This antagonism echoes the logic of science-related populism, defined as an opposition between "(allegedly) virtuous ordinary people" and an "(allegedly) unvirtuous academic elite" that undermines the legitimacy of scientific expertise [12, p.473]. However, not all populist movements are anti-science; some selectively embrace scientific discourse when it aligns with their ideological goals [13].

Despite the growing academic attention to these issues, the field still lacks a comprehensive framework that integrates individual and contextual explanations of conspiracy theory beliefs. Early studies were dominated by psychological approaches, such as the motivational model proposed by Douglas et al. [14], which identifies epistemic, existential, and social motives behind conspiratorial thinking. Yet recent evidence suggests that conspiracy theory beliefs do not necessarily reduce anxiety or uncertainty, and may even reinforce them [15]. Beyond individual psychology, sociological approaches have located conspiracy theory beliefs within broader social transformations associated with the post-truth era: declining institutional trust, inequality, polarization, and fragmented media ecosystems [16,17]. From this perspective, the erosion of confidence in scientific institutions and the contestation of their epistemic authority have become central to understanding SCT.

Because this crisis of trust concerns the relationship between citizens and scientific institutions, the Public Perception of Science framework offers a suitable lens for analysing SCT. Rooted in Science and Technology Studies, this approach examines how knowledge, trust, and engagement shape public attitudes toward science and technology. Yet these dimensions do not emerge in isolation: they are embedded in wider political and institutional contexts that determine how authority and trust are distributed in society. In fact, the very definition of conspiracy theories systematically invokes concepts linked to political science—such as institutions, sovereignty, and anti-elitism—revealing their strong political dimension.

From this standpoint, the quality of democracy may play a crucial role. Democratic systems are characterized by transparency, accountability, and open flows of information, all of which can mitigate the diffusion of unfounded beliefs. Conversely, authoritarian or less democratic contexts tend to foster institutional mistrust and provide fertile ground for conspiratorial thinking.

Therefore, understanding why some citizens are more vulnerable to science-related conspiracy theories requires connecting individual predispositions with the structural environments that shape them. Integrating both levels of analysis allows us to explore not only *who* believes in these theories, but also *under what institutional conditions* such beliefs thrive. This multilevel approach recognizes that attitudes toward science emerge from the interaction between individual dispositions—such as knowledge and trust—and contextual factors, including the political and informational climate of each country.

Building on this perspective, our study draws on three well-established models describing the relationship between science and the public—Scientific Literacy, Public Understanding, and Science-in-Society—each addressing a different dimension of this relationship (knowledge, attitudes, and participation). We integrate these models within a multilevel framework that connects individual dispositions with broader political contexts. Specifically, we test whether the quality of democracy moderates the relationship between citizens' perceptions of science—knowledge, trust, and engagement— and their propensity to believe in SCT.

## Theoretical framework

Since its emergence in the 1960s, the field of Science and Technology Studies has sought to explain how citizens perceive and relate to science. Three main theoretical models have guided this research—Scientific Literacy, Public Understanding, and Science-in-Society [18–20]. Each focus on different dimensions of the relationship between science and the public: knowledge, attitudes, and participation. The aim of this theoretical framework is to examine the results yielded by each approach in studies on unfounded beliefs, particularly conspiracy theories, which are sometimes limited.

The Scientific Literacy model assumes that a higher level of scientific knowledge enhances citizens' ability to evaluate scientific information critically and reduces susceptibility to unfounded beliefs. Early research linked education and factual knowledge to more favourable attitudes toward science [21,22], although the "deficit model" was later questioned for oversimplifying this relationship [20–23].

In the domain of conspiracy theory beliefs, this "knowledge deficit" hypothesis has received partial support. Recent meta-analyses show that scientific literacy and reflective thinking are negatively correlated with conspiracy theory beliefs, even when controlling for sociodemographic characteristics [24]. Nonetheless, other studies suggest that this relationship is indirect and mediated by psychological or social factors such as cognitive style, anxiety, or socioeconomic status [25,26]. Overall, these findings indicate that while knowledge alone does not prevent conspiracy thinking, it provides an essential cognitive resource for critically evaluating misinformation.

H1. Individuals with a higher level of scientific knowledge are less likely to believe in SCT.

The second model, Public Understanding, emphasises the attitudinal dimension of science–society relationship. Here, trust in scientists, perceptions of their integrity, and evaluations of the benefits and risks of scientific progress are key explanatory factors. Research shows that distrust in scientific institutions, rather than ignorance, often drives science skepticism and resistance to expert consensus [18,22,27]. Likewise, perceptions of science as politicized or serving elite interests have been linked to higher endorsement of conspiracy theories [28]. Some studies have even shown that the effect of distrust on conspiracy attribution remains significant regardless of individuals' level of knowledge [29].

H2. Individuals with a more positive attitude towards science show a lower level of belief in SCT

H2.1 Individuals able to recognize the benefits of science show a lower level of belief in SCT.

H2.2 Individuals with greater trust in scientists show a lower level of belief in SCT.

Although scientific literacy and attitudes toward science are conceptually related, they represent distinct dimensions of the public's relationship with science. The former reflects a cognitive component (knowledge of scientific facts and methods), whereas the latter encompasses attitudinal orientations, such as trust in scientists and the perceived benefits of science and technology [20,30,31].

The final model, Science-in-Society, focuses on explaining the subject's relationship with science in terms of the extent to which science is incorporated into aspects of everyday life, such as participation in outreach activities or public engagement in regulatory science. This contextual approach [32–34] explores a more practical dimension of science based on its central role in economic development, public policy and personal life in democratic societies [35]. For example, Lewandowsky et al. [36] showed how people with free-market world views were related to a greater predisposition to reject scientific findings that have potential regulatory implications, such as climate science. In addition, there are studies that have reported the positive effects of bringing the public closer to scientific practice through their participation in scientific research [37–41]. It is therefore to be expected to find an inverse relationship between involvement in the more practical side of science and support for conspiracy theory beliefs. However, no specific literature has been found on this topic, so this study is intended as a first approach to test this type of hypothesis.

H3: Greater engagement in science is associated with reduced belief in SCT.

H3.1 Individuals who are more involved in scientific dissemination activities have a lower level of belief in SCT.

H3.2 Individuals with a more favourable attitude towards involvement in decisions about science and technology have a lower level of belief in SCT.

## Country-level variables in the study of conspiracy theory beliefs

While the three models of public perception of science help explain individual differences in conspiracy theory beliefs, they do not account for the contextual environments in which these perceptions are formed. To address this limitation, we extend the framework to the country level by incorporating political and institutional factors.

The study of structural factors related to the political, cultural and economic context that may encourage belief in conspiracy theories is still in its nascent stages. However, recently relevant voices in the field have begun to urge the study of 1) country-level factors that shape people's willingness to conspiracy theory beliefs and 2) cross-national studies that allow for comparative research [42–44].

There is little but relevant evidence about the influence of cultural factors favouring conspiracy theories acceptance [45–47]. Furthermore, economic factors such as national wealth or economic inequality have also been shown to be significantly related to such beliefs [44,47–49]. However, it is difficult to find papers that test one of the main hypotheses that have sneaked into the academic and popular discourse on conspiracy theories: unfounded beliefs are related to the (poor) condition of Western democracies. Previous work has pointed to the danger that increased conspiracy theory beliefs may generate for the stability of political system and its close relationship with institutional mistrust [50–52]. These hypotheses have rarely been tested and these isolated studies have tended to be based on individual rather than country data [53]. Nevertheless, there is little but relevant evidence about the relation between democratic quality and conspiracy theory beliefs. Hornsey and Pearson [47] have reviewed the available results from various cross-national surveys. In most of them, they found a significant and negative relationship between the level of democracy and the propensity of the population to conspiracy theory beliefs. In our study, we aim to clarify whether this country variable can influence the propensity of European citizens to believe in SCT.

H4. Individuals from countries with a lower level of democratisation are more likely to believe in SCT.

However, in recent years, some work has appeared in the field of conspiracy theories research suggesting that contextual variables can sometimes moderate the relationship between individual variables. Thus, Alper et al. [54] have qualified that the relationship between the level of knowledge and trust in science is positive except in countries with high levels of corruption. In the same year Alper and Imhoff [55] published that the relationship between ideology was a trait that

contributed to explaining individuals' conspiracy theory beliefs only in countries with lower level of corruption, and Bordeleau [42] showed a relationship between political activism and conspiracy theory beliefs, but only in countries that had achieved a certain quality of democracy.

Based on these previous indications, in our study we aim to test whether the quality of democracy in European countries moderates the relationship between the positioning of individuals towards science and belief in SCT. In this way, we will follow the recommendation of Hornsey et al. [56] to make an effort in the area to integrate individual and structural levels of understanding in order to identify eventual moderating effects. From a multilevel perspective, political environments provide the institutional and informational context that shapes how individuals interpret scientific knowledge and authority. Democratic systems, by promoting transparency and open communication, can enhance the efficacy of individual-level factors—such as literacy and trust—in reducing susceptibility to unfounded beliefs.

H5. The relationship between perception of science and belief in SCT is moderated by the level of democratisation of the individual's country.

### Individual control variables

Previous research has shown mixed evidence regarding the role of sociodemographic factors in explaining conspiracy theory beliefs. While some studies have reported associations with gender, age, or socioeconomic status, results are often inconsistent across contexts and types of conspiracy theories [57–65]. Some papers have found a particular relationship between being female and belief in health-related conspiracy theories (vaccines, COVID-19) [57,58]. However, others have showed that men have stronger conspiracy theory beliefs than women have [59]. In any case, what seems clear is that gender becomes more important when it comes to controversial topics [60]. Regarding to age and social class, those papers that have found these variables significant have almost always pointed in the same direction: younger and lower income are characteristics related to conspiracy theory beliefs [61–65]. In this study, age, gender, and economic precariousness were included as control variables, following the most frequent practice in the literature [29,62,61].

Given the multinational nature of the data, we decided not to include educational level as a control variable. The available indicator ("age at which full-time education ended") showed implausible values and required arbitrary recoding across countries. Moreover, since education is strongly correlated with both scientific literacy and attitudes toward science and technology [20,30,31,66]—the core constructs of this study—its inclusion could have introduced collinearity and redundancy.

### Methodology

The information used to test the hypotheses was taken from the Special Eurobarometer [5]. A hybrid survey, face-to-face and online, (N = 37.079) was carried out between April and May 2021. It covers the population of the respective nationalities of the European Union Member States, resident in each of the 27 Member States and aged 15 years and over. The survey has also been conducted in 11 other countries or territories outside the EU: five candidate countries (Albania, Montenegro, North Macedonia, Serbia and Turkey), as well as in Bosnia and Herzegovina, Iceland, Kosovo, Norway, Switzerland and the United Kingdom. To calculate the values of the respondents for each variable, East and West Germany were unified into one country using the "Patch: DE/UK WEIGHT" [67]. In this way, it was also possible to apply the appropriate weighting to the observations.

In Appendix A of the supporting information (see S1 File) the text of the items used to elaborate the necessary variables is provided. The measures and descriptive data are given in Table 1.

### Statistical analysis

Multilevel linear modelling (MLM) was applied, which is appropriate when, as in this case, the data used has a hierarchical structure. In our study, respondents are nested within countries and the number of countries (38) exceeds the minimum value recommended for this type of analysis [68]. This model of analysis allows us to test whether the degree

**Table 1. Measurements and descriptive data for all variables.**

| Variable name and label | Measure (range) | Mean or Percentages | SD |
|---|---|---|---|
| **Dependent variable** | | | |
| **Conspiracy theory beliefs (CONSPI)** | Sum of the values assigned to items 10 and 11 of Q20. Values: 0 = false, 1 = don't know, 2 = True. (0–4) | 1.59 | 1.53 |
| **Variables of the individual's position in relation to science and technology** | | | |
| **Science Literacy** | | | |
| **Science Literacy (SCLITERACY)** | Number of correct answers to items 1–9 of Q20. (0–9) (Cronbach's alpha = 0.678) | 5.32 | 2.02 |
| **Public Understanding** | | | |
| **Mistrust of scientists (MISTRUST)** | Mean of respondent's agreement with item 9 of Q10 and items 1 and 2 of Q11. Values from 1 – Totally disagree- to 5 – Totally agree (1–5) (Cronbach's alpha = 0.669). | 3.30 | 0.88 |
| **Instrumental view of science (INSTRUMENTAL)** | Mean of the respondent's degree of agreement with items 2–4 of Q10. Values from 1 – Totally disagree- to 5 – Totally agree. (1–5) (Cronbach's alpha = 0.602) | 3.40 | 0.82 |
| **Critical view on science (CRITICAL)** | Mean of the respondent's degree of agreement with items 6–8 of Q10. Values from 1 – Totally disagree- to 5 – Totally agree (1–5) (Cronbach's alpha = 0.655). | 3.26 | 0.91 |
| **Blind faith in science (BLINDFAITH)** | Mean of the respondent's degree of agreement with items 7–9 of Q9. Values from 1 – Totally disagree- to 5 – Totally agree (1–5) (Cronbach's alpha = 0.673). | 3.30 | 0.90 |
| **Science-in-Society** | | | |
| **Engagement (ENGAGE)** | Sum of responses to items 1–12 of Q14. Values from 0 – No, never-, to 3, Yes, regularly. Due to the differences in the degree of involvement in the activities, they were weighted, in the same order as in the questionnaire, with weights from 0.1 to 1.2 (0 to 23.40) (Cronbach's alpha = 0.906). | 4.79 | 4.77 |
| **Citizen involvement (INVOLVEM)** | Values 1–4 in the same order of the first four options presented to the respondent (1–4) | 2.35 | 0.72 |
| **Socio-demographic variables** | | | |
| **Sex of respondent (SEX)** | 1 = Woman; 0 = Man | 1 = 51.9% 0 = 48.1% | |
| **Age of respondent (AGE)** | | 48.05 | 17.27 |
| **Economic precariousness (ECONPREC)** | 1 = High precariousness (Highprec); 2 = Low precariousness (Lowprec); 3 = No precariousness (Noprec) | 1 = 6.9% 2 = 25.75% 3 = 67.35% | |
| **Country-level** | | | |
| **Democracy level (DEMO)** | This index published by the Economist Group measures the quality of democracy across the world. It takes values from 1 to 10 and is based on 5 dimensions: electoral process and pluralism, functioning of government, political participation, political culture and civil liberties. (1.1 to 8.8) | 6.64 | 2.0 |

of conspiracy theory belief is determined both by individual characteristics (level 1 variables) and by contextual variables specific to the country where they reside (level 2 variables). In addition, it offers the possibility of assessing the interaction between variables at both levels to establish whether different contexts produce differences in individual relationships with the degree of conspiracy theory belief.

The individual's independent variables were centred by the country mean and the DEMO variable was centred by the grand mean, since, in order to achieve the objective outlined in this study, there is interest in probing all the effects described above [69–71].

The hierarchical analysis process started with the empty model (model 0) to check the Intraclass Correlation Coefficient (ICC). This indicator yielded a Design Effect value of 231.5, which makes the application of MLM advisable since it is well above the minimum value required [70,72]. Next, all individual variables (model 1) and then the level 2 variable DEMO

(model 2) were included in the equation as fixed effects. The next step also contained the interaction between the country variable and two level 1 variables, SCLITERACY and MISTRUST (model 3). Finally, these last two factors were included separately as random effects (models 4a and 4b).

The different models produced in the process of variable inclusion were compared. On the one hand, the likelihood ratio test was used to evaluate nested models and to check whether the inclusion of the additional parameters produced a significantly better fit. For this purpose, the equations were estimated using full information maximum likelihood (FIML), following the recommendation of Peugh [70], Aguinis et al. [73] and Tabachnick and Fidell [74 p. 834], since this estimation yields deviance values that take into account both the coefficients and the variance components. On the other hand, the proportional reduction in variance (PRV) was also calculated as a measure of local effect size [70].

## Results

Table 2 shows the results of the multilevel linear modelling. Based on the values shown in the empty model (model 0), the between-country variability is 0.559 and the within-country variability is 1.805. These results produce the aforementioned ICC value equal to 0.236, indicating that 23.6% of the total variance of CONSPI can be attributed to differences between countries.

Examining the parameters of the fixed effects, we find that they remain virtually unchanged from the time the variable is added until the end of the hierarchical inclusion process. Model 1 includes all the individual variables related to each of the theoretical frameworks of public perception of science, and the control variables (age, sex and economic precariousness). Before analysing how each variable is influenced by the parameters obtained as fixed effects, it is interesting to know the size of the effect of each of the three separate perception models on the belief SCT. In Appendix B of the supporting information (see S1 File) it can be seen that the model that produces the largest reduction of intra-country variance is the Public Understanding model (10.2%), which includes variables related to attitudes towards science. The Science Literacy and Science in Society models produce a much smaller reduction (5.8% and 0.8% respectively).

The effect produced by the variable SCLITERACY is negative and significant, thus confirming the approach proposed in H1. In general, the formulations in H2 are also supported by the data: those who distrust scientists more (MISTRUST) have a higher level of conspiracy theory belief (H2.2). As will be seen below, this variable reduces a larger percentage of the intra-country variance, thus positioning itself as the science perception characteristic that contributes most to conspiracy thinking formation. The negative effect of INSTRUMENTAL and the positive effect of CRITICAL on the explained variable is also as expected (H2.1). Those people who are more likely to identify the harms of science and technology than their potential benefits are more likely to believe in SCT. Surprisingly, the BLINDFAITH variable indicates that these people have an overly optimistic (even naive) view of science in terms of what science is capable of achieving. With respect to the variables that make up the Science-in-Society model, the results show that those who are more involved in science and technology-related issues (ENGAGE) tend to believe less in SCT (H3.1). On the other hand, those who advocate the democratisation of science and consider it necessary for citizens to participate in decision-making in science (INVOLVEM) also seem more likely to believe in conspiracies theories within the same field (H3.2). Therefore, H3 is only partially confirmed.

In relation to the sociodemographic variables that we included as control variables, we found that the type of conspiracy theories studied in this study are not associated with the gender of the individuals (SEX). On the other hand, age (AGE) shows a significant and negative relationship with our dependent variable. Younger people tend to have a higher degree of conspiracy theory belief. Finally, the economic situation of households is also significant in the model. Those who are in a precarious economic situation, regardless of the degree (Highprec, Lowprec), are more likely to conspiracy theory beliefs of a scientific nature than those who do not have problems paying their bills (Noprec).

In model 2 we add a country-level variable (DEMO) that expresses the level of democratic quality of a country. This factor has a clear negative effect, showing that the inhabitants of countries with higher levels of democracy have lower

**Table 2. Parameters of the multilevel regression analysis, standard error in parentheses.**

| | Model 0 | Model 1 | Model 2 | Model 3 | Model 4a | Model 4b |
|---|---|---|---|---|---|---|
| **Fixed effects** | | | | | | |
| Intercept | 1.616 (0.121)*** | 1.619 (0.122)*** | 1.607 (0.075)*** | 1.607 (0.075)*** | 1.607 (0.075)*** | 1.607 (0.075)*** |
| SCLITERACY | | −0.133 (0.004)*** | −0.133 (0.004)*** | −0.133 (0.004)*** | −0.130 (0.012)*** | −0.132 (0.004)*** |
| MISTRUST | | 0.347 (0.009)*** | 0.347 (0.009)*** | 0.342 (0.009)*** | 0.340 (0.009)*** | 0.346 (0.023)*** |
| INSTRUMENTAL | | −0.094 (0.010)*** | −0.094 (0.010)*** | −0.096 (0.010)*** | −0.095 (0.010)*** | −0.093 (0.010)*** |
| CRITICAL | | 0.149 (0.010)*** | 0.149 (0.010)*** | 0.145 (0.010)*** | 0.139 (0.010)*** | 0.144 (0.010)*** |
| BLINDFAITH | | 0.091 (0.009)*** | 0.091 (0.009)*** | 0.088 (0.009)*** | 0.084 (0.009)*** | 0.089 (0.009)*** |
| ENGAGE | | −0.012 (0.001)*** | −0.012 (0.001)*** | −0.011 (0.001)*** | −0.011 (0.001)*** | −0.011 (0.001)*** |
| INVOLVEM | | 0.092 (0.009)*** | 0.092 (0.009)*** | 0.093 (0.009)*** | 0.091 (0.009)*** | 0.092 (0.009)*** |
| SEX | | 0.020 (0.013) | 0.020 (0.013) | 0.013 (0.013) | 0.014 (0.013) | 0.015 (0.013) |
| AGE | | −0.004 (<0.001)*** | −0.004 (<0.001)*** | −0.004 (<0.001)*** | −0.004 (<0.001)*** | −0.004 (<0.001)*** |
| Highprec[1] | | 0.398 (0.028)*** | 0.398 (0.028)*** | 0.405 (0.028)*** | 0.410 (0.028)*** | 0.392 (0.028)*** |
| Lowprec[1] | | 0.265 (0.017)*** | 0.265 (0.017)*** | 0.267 (0.017)*** | 0.262 (0.017)*** | 0.262 (0.017)*** |
| DEMO | | | −0.299 (0.038)*** | −0.298 (0.038)*** | −0.299 (0.038)*** | −0.299 (0.038)*** |
| SCLITERACY* DEMO | | | | −0.019 (0.002)*** | −0.022 (0.006)*** | −0.020 (0.002)*** |
| MISTRUST * DEMO | | | | −0.008 (0.004) | −0.002 (0.004) | 0.003 (0.011) |
| **Random effects** | | | | | | |
| Residual | 1.805 (0.013)*** | 1.533 (0.011)*** | 1.533 (0.011)*** | 1.528 (0.011)*** | 1.513 (0.011)*** | 1.516 (0.011)*** |
| Intercept | 0.559 (0.131)*** | 0.565 (0.130)*** | 0.214 (0.050)*** | 0.215 (0.050)*** | 0.213 (0.049)*** | 0.214 (0.050)*** |
| SCLITERACY UN(2,2) | | | | | 0.005 (0.001)*** | |
| MISTRUST UN(2,2) | | | | | | 0.016 (0.004)*** |
| **Elements of model fits** | | | | | | |
| −2 Log Likelihood | 127335,87 | 118202,45 | 118165,78 | 118055,29 | 117784,78 | 117856,53 |
| Number of Parameters | 3 | 14 | 15 | 17 | 19 | 19 |

*** p<0.001.

(1) No precariousness (Noprec) is the reference.

levels of conspiracy theory beliefs (H4). The rest of the variables in the model are not affected in their relationship with the dependent variable and maintain similar levels of significance and directionality in the rest of the models in the analysis. Fig 1 shows in detail the positioning of the countries included in the study in relation to the CONSPI and DEMO values, as

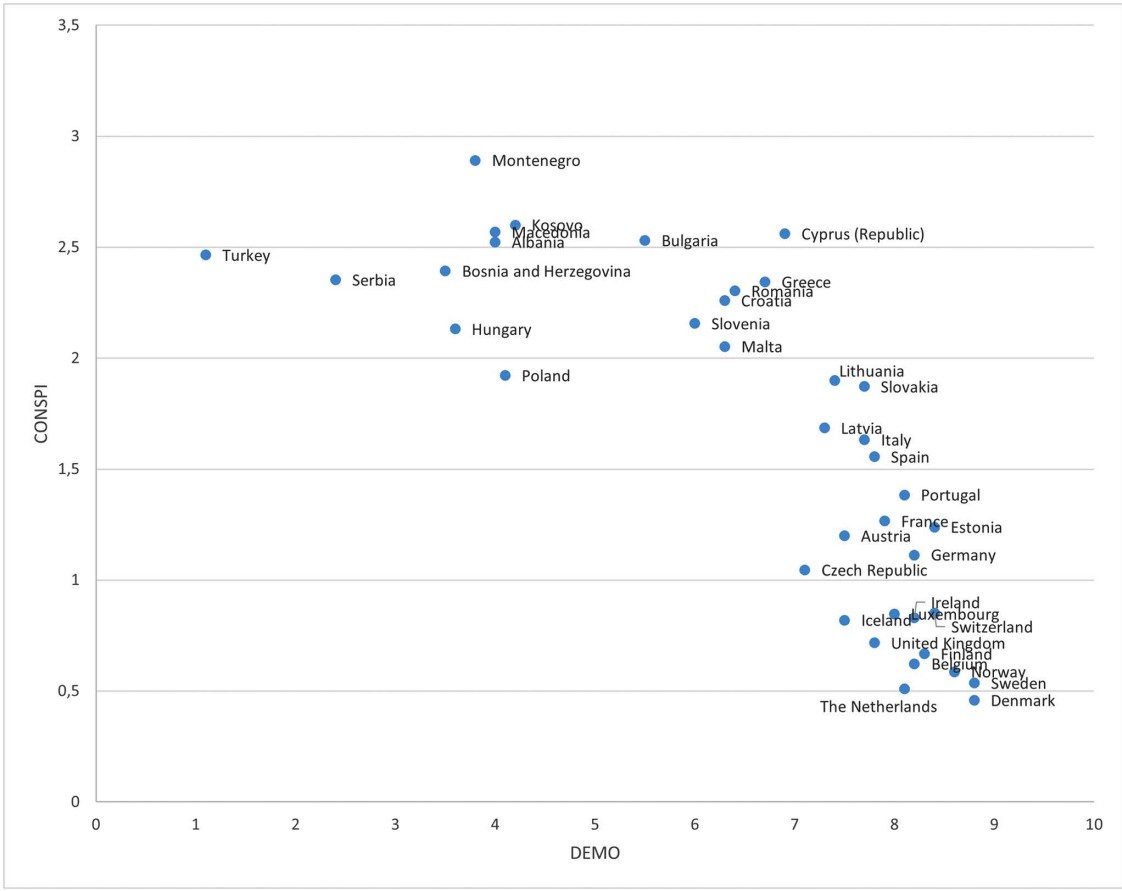

**Fig 1. Relation between conspiracy theory beliefs and democracy index.**

well as a nearly linear relationship between both variables. The lower right corner is occupied by fully established European democracies such as those of the Scandinavian countries and the UK. In the middle of the graph we find Mediterranean countries and some former Soviet republics (those that became independent earlier) while in the upper left corner are fundamentally the republics that were part of the former Yugoslavia, plus Turkey and Hungary. It should be noted that SCT belief decreases sharply only when countries reach a real democratic status (above 6).

The summary presented in Table 3, shows that as we move up the hierarchy followed in our analysis each model fits the data better than the previous one, as all differences in the deviance values, $(−2LogL_{Reduced\ model}) -(−2LogL_{Full\ model})$, are significant. It can also be seen that the inclusion of all individual variables, model 1, gives rise to a PRV of the residual variance of 15.1%, indicating that the respondent characteristics included in the equation contribute to explaining that percentage of the level of within-country conspiracy theory beliefs. This set of factors, on the other hand, generates more between-country variance. In contrast, when adding the country democracy index, model 2, the between-country variance decreases significantly (62.1%). This reduction indicates that, of the total CONSPI variance attributed to differences between countries, 14.7% (0.236x0.621) is explained by the DEMO variable.

This suggests that the quality of democracy may moderate the relation between some individual variables, which justifies cross-level interactions [69]. As discussed in the theoretical framework, the socio-political context may determine the relationship between different elements of individuals' positioning towards science. It is therefore of substantive interest to test

**Table 3. Multilevel effect size reporting.**

| Comparative models | Likelihood Ratio Model Testing $\chi^2= (-2LogL_{Reduced\ model})- (-2LogL_{Full\ model})$ | Multilevel Effect Size Reporting | |
|---|---|---|---|
| | | PRV (residual variance) | PRV (intercept variance) |
| **M1-M0** | $\chi^2$ (11)=9133.42(p<0.001) | 15.1 | −1.1[(1)] |
| **M2-M1** | $\chi^2$ (1)=36.67(p<0.001) | 0 | 62.1 |
| **M3-M2** | $\chi^2$ (2)=110.49(p<0.001) | 0.03 | −0.5[(1)] |
| **M4a-M3** | $\chi^2$ (2)=270.51(p<0.001) | 1.0 | 0.9 |
| **M4b-M3** | $\chi^2$ (2)=198.76(p<0.001) | 0.8 | 0.5 |

[(1)]Local (model with less IV) effect sizes that exceed global (model with more IV) effect sizes are possible in multilevel analyses [75].

such interactions with science perception variables included in the model [73]. SCLITERACY and MISTRUST were selected as lower-level variables because they turned out to be the two variables that, individually, produced the largest PRV in the within-country variance against the empty model, 5.8% and 9% respectively (see Appendix C in S1 File). The addition of the interaction (Model 3, Table 2) produced a significant improvement in model fit ($\chi^2$(2) =110.49; p<0.001). Thus, although the value of the PRV was negligible, the two lower-level variables were included as random effects [73] separately [74: 833]. The latter two models (4a and 4b in Table 2) also yield a significantly better fit than the model without random slopes.

The results show a negative effect on the interaction between democracy and knowledge and a variance of the slopes that is significantly different from zero (Model 4a). This indicates that higher levels of democracy intensify the negative relationship between scientific literacy and belief in SCT. On the other hand, although no significant cross-level interaction was found between democracy and distrust, non-zero variance was also obtained when incorporating this attitudinal item as a random effect (Model 4b). These results allow us to partially confirm our H5, that is, the moderating effect of the level of democracy only occurs in the relationship between knowledge and belief in SCT.

In order to graphically visualise the nature and direction of the interactions and random effects described [73] the graphs presented in Figs 2 and 3 were produced. To do so, countries were grouped according to their level of democratic quality following the typology proposed by the Economist Intelligence Unit: Full democracies (8–10), Flawed democracies (6–7.99), Hybrid regimes (4–5.99), and Authoritarian regimes (0–3.99) (Appendix D in S1 File shows which group each country used in the study falls into).

The first figure shows that Hybrid and Authoritarian regimes have a low slope, so these are countries where increased knowledge hardly reduces the level of conspiracy theory belief. The opposite is true for countries with high rates of democracy. In these cases, the slope is high and, in addition to starting with lower levels of conspiracy theory beliefs for the lowest values of SCLITERACY, it even reaches a zero value of conspiracy when the degree of knowledge reaches maximum values.

In Fig 3 it is interesting to see that countries with low levels of democracy have a relationship between distrust in scientists and belief in SCT that is always higher than that observed in countries with greater democratic guarantees. In fact, we see that in these countries, even at lower levels of distrust, some level of conspiracy theory belief occurs and it reaches the highest level when distrust increases. In full democracies, however, the level of conspiracy thinking is only moderate for the most extreme scores of distrust towards scientists.

## Discussion

The Science and Technology Studies approach can help to explain the public's belief in SCT and provide some evidence to address the concern raised by Harambam [76] about the lack of discussion of this topic by scholars in this field. Of the

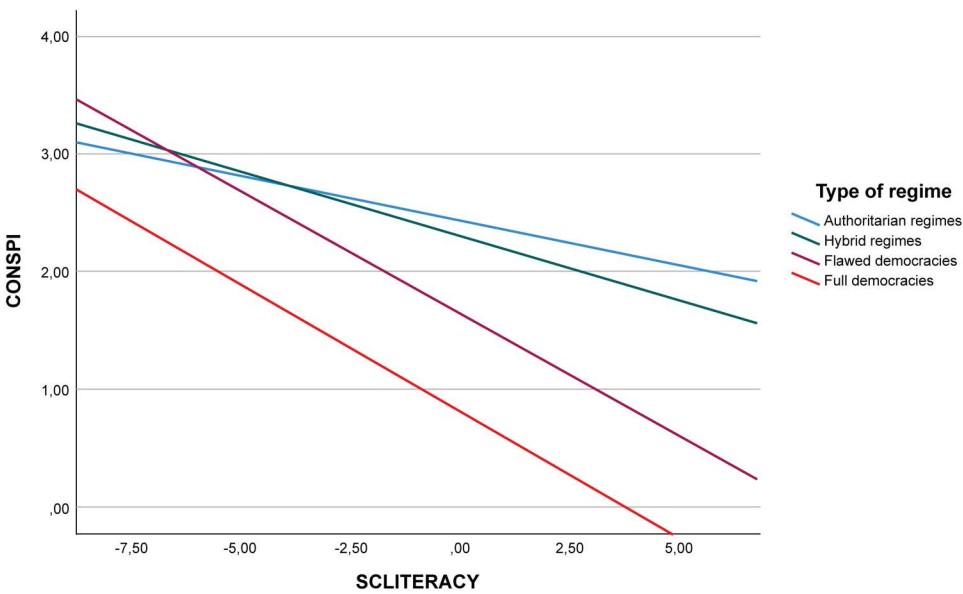

**Fig 2. Relationship between CONSPI and SCLITERACY according to the level of democracy.**

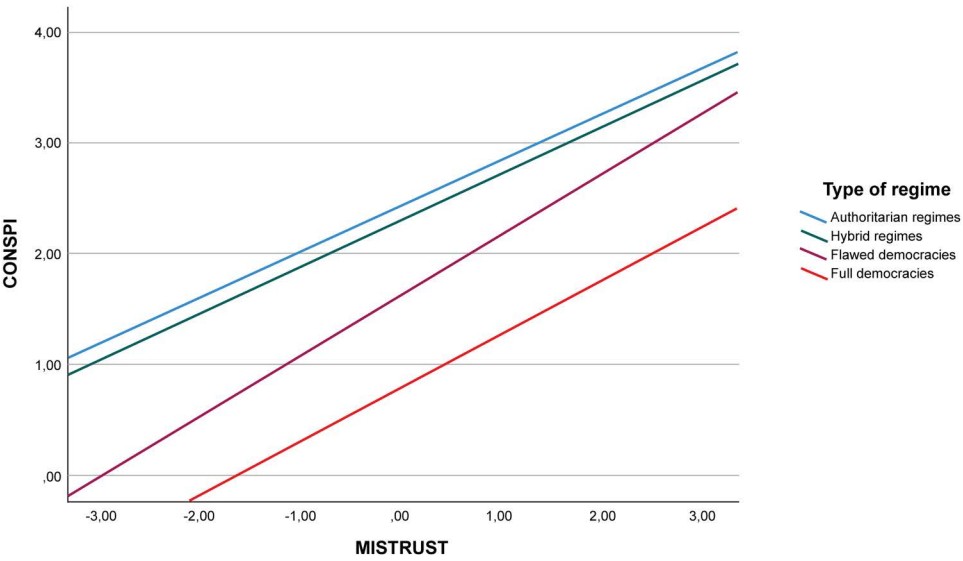

**Fig 3. Relationship between CONSPI and MISTRUST according to the level of democracy.**

various factors influencing public perception of science, a lack of trust in scientists is the most significant contributor to the acceptance of unwarranted beliefs, followed by the level of scientific knowledge and a critical attitude towards science. Therefore, of the three theories of public perception of science, Public Understanding contributes the most to explaining this fact.

This contrasts with the traditional emphasis in the field of science communication to combat misinformation or negative attitudes towards science (e.g., vaccines, health measures and technological innovations) by promoting scientific literacy.

However, when it comes to controversial scientific issues, educational level or scientific knowledge does not significantly impact the likelihood of acting in accordance with scientific consensus [60,77]. The idea that making facts available to the public contributes to changing minds has been widely contested [78]. Indeed, in some cases a 'boomerang effect' has been observed, with public polarisation increasing in response to science-based messages related to climate change [79]. Nevertheless, our findings align with previous literature which indicates that education is negatively correlated with conspiracy theory beliefs, although it is not the most significant factor in explaining this phenomenon [80,57].

Attitudinal variables, especially distrust of scientists and a critical or sceptical view of how science works, play a key role in explaining belief in SCT. Our results are in line with other studies that have shown that the rise of this type of theories may be related to a problematic relationship with scientific authority in a context of growing institutional distrust [81–85]. Conspiracy theory beliefs, therefore, would have less to do with a public discussion about the accuracy of scientific facts than with a much deeper phenomenon: the dissolution of a shared truth and the questioning of epistemological sources on which there used to be consensus (the epistemic trust pointed by Vaupotič et al. [86]. Indeed, Nefes et al. [81] have recently shown how a spectrum of communities dedicated to the spread of conspiracy theories emerged in the public conversation on X about distrust of political and health institutions during the COVID-19 pandemic. Distrust was the catalyst for the spread of conspiracy theories and the "gateway" connecting one to the other.

Our results also are aligned with the theory proposed by Mede and Schafer [12] about scientific populism. This phenomenon is characterised by the transposition of the anti-elitist discourse to the field of science and the antagonism between a virtuous ordinary people and an unvirtuous academic elite represented by scientist and scholarly institutions. This discursive framework may have important consequences for the gradual loss of legitimacy of scientific institutions and their epistemic authority.

However, there is an important nuance to this breakdown in trust. Our results show that those who believe in SCT tend to distrust scientists and have a critical view of how science works. But, at the same time, they have very high — albeit misguided — expectations of what science can achieve. Indeed, some studies have shown that people who believe in conspiracy theories are at least partly pro-science [87]. Nevertheless, this kind of "naive" blind faith in science as a system of knowledge suggests that the trust at stake is not in the scientific method, but in the potential corruption of those who implement it, thus fostering the idea of the unvirtuous academic elite described by Mede and Schafer [12].

The relatively low explanatory power of the *Science-in-Society* model suggests that attitudes promoting the democratisation of science and public participation may not directly translate into lower conspiracy theory belief. These attitudes often represent normative views about how science *should* operate, rather than perceptions of how it *actually* works in practice. Consequently, their influence might be more indirect and context-dependent. For example, in contexts where institutional trust is low, calls for greater participation could coexist with scepticism or disillusionment, thereby limiting their protective effect against conspiracy theory beliefs.

However, it is interesting to discuss the apparently contradictory direction taken by the two variables included in this model. On the one hand, greater engagement in science-related activities — such as visiting museums or attending public lectures — is associated with lower conspiracy theory belief. On the other hand, individuals who endorse conspiracy theories express stronger support for public participation in science and technology decision-making. While counterintuitive at first, this finding may reflect a broader tension between epistemic distrust and demands for participatory inclusion.

This paradox resonates with the growing literature on science-related populism and anti-expert sentiment, which interprets such attitudes as a reaction to perceived exclusion by scientific elites rather than a rejection of science itself [12,88,89]. People who believe in conspiracy theories may distrust scientists and hold critical views about how science operates, while simultaneously maintaining high — albeit sometimes unrealistic — expectations of what science can achieve. Previous studies have indeed shown that conspiracy believers are, at least partly, pro-science [87], suggesting that their distrust lies not in the epistemic system but in those who administer it. This helps explain their desire for more citizen involvement in science governance — an attempt to "reclaim" control from perceived corrupt or self-serving elites.

                                                                    

Recent scholarship also provides mixed evidence on the link between conspiracy theory beliefs and civic or political engagement. While Jolley and Douglas [90] speculate that conspiracy theory beliefs may suppress collective action and participation, Kim [67] finds that they can, in some contexts, stimulate engagement. In fact, in recent years some findings have clarified the role of activism in relation to conspiracy theories. People who believe in such theories are actually more prone to nonnormative political engagement [91] and online activism [92]. Bordeleau et al. [75], in turn, note that this relationship between activism and conspiracy theory beliefs occurs only in consolidated democracies, but not necessarily in developing democracies or more authoritarian regimes. Our findings contribute to this debate by suggesting that, in the scientific domain, conspiracy theory beliefs may foster a reactive form of engagement — one driven less by democratic ideals than by perceived alienation from expert decision-making. This interpretation situates our results within broader discussions about science populism, participatory democracy, and the social consequences of declining epistemic trust.

According to our results, people who believe in SCT tend to be critical of the system, particularly the way science is conducted. This attitude of dissatisfaction is often accompanied by a more generalised resentment, which can lead to a sense of alienation due to the perceived lack of opportunities to participate in public life [93,94]. When this social discontent is combined with economic disadvantage, it fosters a climate in which conspiracy theories are more likely to be embraced [49]. Therefore, it is not surprising that economic precariousness, among the control variables included in our study, has the greatest influence on the likelihood of believing in SCT.

Many of these questions suggest that a mismatch between the subject and their social, political and economic environment could trigger conspiracy theory beliefs. In line with these findings, recent work has emphasised the importance of considering contextual factors when exploring the extent to which these may contribute to an increase in conspiracies acceptance. In recent years, aspects such as wealth and its distribution, as well as the quality of democratic systems, have become increasingly important. However, Hornsey and Pearson [47] suggest that the results may still differ depending on how conspiracy theories were measured, with the potential for inconsistencies in the findings. Our analysis revealed a strong correlation between a country's democratic quality and the prevalence of belief in SCT among its population. In line with the findings of Hornsey and Pearson's review [47], conspiracy theorizing tended to be higher among more authoritarian nations, although this effect did not emerge reliably in the largest of their datasets. However, this association must be interpreted cautiously, given the strong collinearity between democracy and other structural factors such as economic development, institutional effectiveness, and corruption control. Future research using disaggregated or multi-indicator models could clarify the extent to which these effects are distinct or mutually reinforcing. We emphasise that more research is needed to confirm this initial impression of the relationship between SCT and trust-sensitive political realities.

This approach has become to shift the paradigm by moving conspiracy theory beliefs away from the mindset of individuals, which is influenced by personal characteristics, towards a more complex, context-dependent understanding of the phenomenon. This does not mean disregarding individual variables, but rather seeking correlations with social variables at the country level in order to gain a more comprehensive understanding of the SCT phenomenon. As Hornsey et al. [56] points out, the challenge in this area is to test for potential 'ecological fallacies' that could be affecting the evidence about conspiracy theory beliefs provided by individual variables so far.

Nonetheless, there is still a long way to go in efforts to capture cross-national differences in conspiracy theory beliefs. One of the most promising lines of analysis consist in the articulation of individual and contextual variables for a better understanding of SCT acceptance process. For example, de Boer and Aiking [48] found that the positive attitude toward vaccines had a stronger effect on opposing conspiracy beliefs in higher-income countries than in lower-income countries. In this regard, one of our most notable achievements in this study is demonstrating the moderating effect of the democracy index on individual variables acting as protective factor against SCT. Thus, citizens of countries with high levels of democratic quality have lower levels of SCT beliefs even among individuals with low scientific literacy or high distrust of scientists. Moreover, the negative relationship between scientific literacy and SCT beliefs is especially pronounced in

countries with high democratic quality. In contrast, in countries classified as Hybrid and Authoritarian regimes, the proportion of citizens who believe in SCT remains high, with only a slight decrease even among those with the highest levels of scientific literacy. This fact may be related to the exception found by Alper et al. [54] in the relationship between knowledge and conspiracy theory beliefs: this relationship is always negative except for those individuals residing in countries with high levels of corruption.

One possible explanation for this asymmetric moderating effect lies in the different nature of the two predictors. Scientific literacy depends heavily on contextual factors such as educational systems, transparency, and access to reliable information — all of which tend to be more robust in consolidated democracies. Therefore, democratic quality can amplify the protective role of scientific knowledge by fostering environments where citizens are better equipped to evaluate evidence, contrast sources, and resist misinformation. In contrast, distrust of scientists operates through a different mechanism. It is often driven by affective, ideological, or identity-based orientations that are less sensitive to institutional conditions and more associated with cultural or populist narratives about expertise and power. Such distrust can persist, or even intensify, in democratic contexts, where pluralism and freedom of expression allow anti-elitist and conspiratorial discourses to circulate more openly. Consequently, democracy moderates the effect of knowledge but not that of distrust, since the former is shaped by structural and informational resources, whereas the latter reflects deeper cultural and value-based worldviews.

Taken together, these findings show that the determinants of science-related conspiracy theory beliefs operate at multiple levels. Individual predispositions—such as knowledge and trust—shape the likelihood of endorsing these beliefs, but their influence depends on the broader institutional context. The quality of democracy contributes to defining this context by establishing the levels of transparency, pluralism, and accountability within which citizens evaluate scientific information. Thus, democratic environments not only strengthen public trust in science but also enhance the capacity of individual factors to counter misinformation and unfounded beliefs.

Due to the embryonic state of the study of the relationship between structural variables and predisposition towards conspiracy theory beliefs, future research in this area is promising. More research effort is needed to clarify the role played by economic, political, and cultural factors in individual SCT acceptance. However, this approach must face an important limitation because such national variables often exhibit high collinearity which will require careful statistical analysis. To address this limitation, we specified a parsimonious Level-2 model including only the Democracy Index as the country-level predictor. The observed negative association between democracy and belief in science-related conspiracy theories should therefore be interpreted as a robust cross-national correlation that may partly reflect other correlated macro characteristics (e.g., GDP per capita, corruption, or government effectiveness), rather than an isolated causal effect of democracy.

Moreover, our findings regarding blind faith in science highlight the need to refine how different forms of trust in science are conceptualised and measured. Future studies could differentiate between trust in scientific methods, institutions, and actors, as well as explore how naive or uncritical faith in science manifests behaviorally — for instance, through the unreserved acceptance of scientific claims or the expectation that science can provide solutions to all societal challenges.

Finally, this new line of research on SCT could help to improve the effectiveness of European public policies aimed at tackling misinformation. Traditionally, these types of measures have focused on the importance of knowledge and scientific literacy in addressing unfounded beliefs. However, initial evidence suggests that there is no universal 'recipe' for reducing conspiracy theory beliefs, and that these types of policies must be adapted to each context to be impactful.

## Supporting information

**S1 File. Variables information and supplemental analyses.**
(PDF)

## Acknowledgments

We are grateful to Professor Alberto del Rey for his valuable comments and suggestions, which substantially improved the analysis of this work.

## Author contributions

**Conceptualization:** Irene López-Navarro, Libia Santos-Requejo.

**Data curation:** Irene López-Navarro, Libia Santos-Requejo.

**Formal analysis:** Irene López-Navarro, Libia Santos-Requejo.

**Investigation:** Irene López-Navarro, Libia Santos-Requejo.

**Methodology:** Irene López-Navarro, Libia Santos-Requejo.

**Visualization:** Irene López-Navarro, Libia Santos-Requejo.

**Writing – original draft:** Irene López-Navarro, Libia Santos-Requejo.

**Writing – review & editing:** Irene López-Navarro, Libia Santos-Requejo.

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
