## [Decision Letter · Decision Letter 0]

7 Sep 2025

Dear Dr. López-Navarro,

We look forward to receiving your revised manuscript.

Kind regards,

Cengiz Erisen

Academic Editor

PLOS ONE

Journal Requirements:

“This paper is part of the project ‘The role of distributed and dialogic expertise in the resolution of public scientific-technological controversies: an epistemological, argumentative and sociological analysis’. It has been funded by the Spanish Ministry of Science and Innovation. Code PID2019-105783 GB-I00.”

Reviewers' comments:

Reviewer's Responses to Questions

1. Is the manuscript technically sound, and do the data support the conclusions?

Reviewer #1: Yes

Reviewer #2: Yes

Reviewer #3: Yes

2. Has the statistical analysis been performed appropriately and rigorously?

Reviewer #1: Yes

Reviewer #2: Yes

Reviewer #3: Yes

3. Have the authors made all data underlying the findings in their manuscript fully available?

Reviewer #1: Yes

Reviewer #2: Yes

Reviewer #3: Yes

4. Is the manuscript presented in an intelligible fashion and written in standard English?

Reviewer #1: Yes

Reviewer #2: Yes

Reviewer #3: Yes

Reviewer #1: This research article investigates which factors are associated with belief in science-related conspiracy theories (SCTs) across Europe. Using Eurobarometer survey data from over 37,000 participants, the authors examine how scientific literacy, attitudes toward science, and the democratic quality of a country relate to SCT beliefs at both individual and national levels.

The primary aim is to test three established models of public understanding of science: Science Literacy, Public Understanding, and Science-in-Society by analyzing their links with SCT belief. In addition, the study asks whether a country’s democratic quality shapes these associations. The authors use cross-sectional survey data and multi-level regression modeling to explore these questions.

Main Hypotheses and Key Findings

• H1: People with higher scientific knowledge are less likely to believe in science-related conspiracy theories (SCTs).

Finding: The data support this. The research showed a negative and significant relationship between scientific literacy (SCLITERACY) and belief in SCTs. However, this link is much stronger in countries with higher democracy. In authoritarian or hybrid regimes, an increase in scientific knowledge leads to only a minimal reduction in conspiracy beliefs. In full democracies, scientific knowledge makes a real difference, sometimes nearly eliminating SCT belief.

• H2: People with a more positive attitude toward science are less likely to believe in SCTs.

H2.1: Seeing more benefits in science predicts lower SCT belief.

H2.2: Trusting scientists predicts lower SCT belief.

Finding: These are confirmed, especially for trust. Mistrust in scientists is the most potent driver of conspiracy belief, much more than simply lacking knowledge. Incorporating these attitudinal variables, the "Public Understanding" model explained the most significant proportion of intra-country variance (10.2%), highlighting its strong explanatory power. This suggests that the rise of these theories is deeply connected to a strained relationship with scientific authority and growing institutional distrust. The study also found that individuals who were more prone to identifying the potential harms of science and technology (CRITICAL view) were more likely to believe in SCTs. Surprisingly, those with an overly optimistic or "blind faith" view of science (BLINDFAITH) also showed a higher tendency to believe in SCTs.

• H3: People more involved with science will believe less in SCTs.

H3.1: Taking part in science-related activities is linked to lower SCT beliefs.

H3.2: Favoring public involvement in science decision-making should predict lower SCT belief.

Finding: There is partial support. Being active in science (like following popular science or participating in citizen science) is linked to less belief in conspiracy theories. But surprisingly, people who want more lay involvement in science policy have higher conspiracy beliefs.

• H4: People in less democratic countries are more likely to believe in SCTs.

Finding: This is strongly supported. Living in a democracy lowers the odds of believing in science-related conspiracies. Once a country passes a certain threshold for democracy, SCT belief drops sharply.

• H5: The relationship between attitudes/knowledge about science and conspiracy belief is influenced by how democratic a country is.

Finding: The moderating effect of democracy is only observed in the case of scientific knowledge. Democracy strengthens the negative link between knowledge and conspiracy belief, making knowledge more “protective” in democratic countries. For other attitudes (like trust), this interaction is not significant.

Other Findings (Control Variables):

• Age: Younger people are more likely to believe in SCTs.

• Economic hardship: Financial difficulties are linked to greater conspiracy belief.

• Gender: No significant effect on this dataset.

In short, this study shows that belief in science-related conspiracy theories cannot be explained by lack of knowledge alone. The most critical factors are trust in scientists and the broader democratic context. Living in a democratic society reduces SCT belief, even among those with less knowledge or higher mistrust. The findings strongly suggest that public policies aimed at combating misinformation should be tailored to specific national contexts, rather than relying on a one-size-fits-all approach that primarily emphasizes knowledge dissemination.

Given the study's strengths, especially its robust methodology, the breadth of its dataset, and its thoughtful analytical approach, I believe this manuscript makes a valuable contribution to the literature on science-related conspiracy beliefs. With a few clarifications and minor improvements as suggested below, the article would be well-suited for publication in PLOS ONE. Addressing these points will help further strengthen the clarity and practical value of the paper.

Key Strengths:

1- Addressing a Critical and Timely Issue:

This paper addresses the significant issue of conspiracy theories related to science, which have substantial implications for both individual and public health. The research feels particularly relevant given today’s climate of misinformation and growing public concern about the impact of these beliefs.

2- Having a Robust Methodology and a Large Dataset:

The paper uses data from a Special Eurobarometer survey, with 37,079 participants drawn from 27 EU countries and 11 other countries or territories. This large and diverse sample makes the findings generalizable and statistically solid. Multilevel Linear Modelling (MLM) fits the nested structure of the data, individuals within countries, and lets the authors analyze individual and national factors and their interactions. The ICC of 0.236 shows that nearly a quarter of the variation in conspiracy beliefs comes from country-level differences, which confirms that MLM is appropriate.

3- Challenging the “Deficit Model” and Offering a Nuanced Perspective:

One of the article’s central contributions is challenging the standard idea that believing in conspiracy theories is solely a lack of knowledge. The authors show that, although scientific literacy is linked to lower conspiracy beliefs, mistrust in scientists is a more decisive factor. They also highlight the protective effect of democracy at the national level, moving beyond the traditional “deficit model”.

4- Integrating Established Theoretical Frameworks:

The research paper is notable for systematically testing three major models from literature: Science Literacy, Public Understanding, and Science-in-Society. This comprehensive approach gives a nuanced picture of why people accept or reject science-related conspiracy theories.

5- Offering Innovative Focus on Country-Level Factors:

Another key strength is the study examining how democracy moderates the relationship between science perceptions and conspiracy beliefs. This focus on national context is original. The findings show that democracy acts as a buffer: people in more democratic countries are less likely to believe in SCTs, even when their trust in scientists is low or their scientific knowledge is limited.

6- Highlighting Practical Policy Implications:

The results are directly relevant for policymakers. The authors show that effective responses to misinformation can’t be one-size-fits-all. Policy needs to be tailored to each country’s context, not just focused on increasing knowledge.

7- Identifying Key Factors:

Finally, the study indicates that mistrust of scientists is the primary factor driving belief in science-related conspiracy theories, which account for the most significant proportion of within-country variation.

Areas for Improvement

1- Partial Confirmation of Hypotheses

1.1 Hypothesis 3 (Engagement in Science): This hypothesis received only partial support. Engagement in scientific dissemination activities (ENGAGE) showed the expected negative correlation with SCT belief, but, interestingly, a favorable attitude towards citizen involvement in science decision-making (INVOLVEM) was linked to higher belief in SCTs.

Suggested improvement: Expanding the discussion around this unexpected result would be beneficial. The authors interpret this pattern as reflecting “alternative epistemologies” and dissatisfaction with academic elites. However, further theoretical exploration and/or qualitative research could clarify why greater support for lay involvement is associated with increased conspiracy belief. The authors may consider strengthening this part of the discussion by referencing the growing literature on science populism and anti-expert sentiment to clarify whether this association reflects a genuine demand for participatory democracy or, rather, a reaction against elites. It would be beneficial to cite recent studies and deepen the interpretation, enriching the discussion and situating the findings within broader debates.

In addition, recent scholarship shows a growing debate on the link between conspiracy beliefs and civic or political engagement. While some studies, such as Jolley and Douglas (2014) argue that conspiracy beliefs may suppress participation, others, like Kim (2022), find that conspiracy beliefs can stimulate engagement. In this paper, the unexpected positive association between support for citizen involvement in science decision-making and SCT belief (H3) could be better supported by directly referencing these contrasting perspectives in the literature. This would help readers understand such findings. However, at first counterintuitive, it aligns with emerging evidence that conspiracy beliefs sometimes motivate individuals to seek greater influence and involvement, often as a reaction to perceived exclusion by experts or institutions.

1.2 Hypothesis 5 (Moderation Effect): This hypothesis was only partially supported. The moderating effect of democracy was apparent in the relationship between scientific knowledge and SCT belief, but not in the link between distrust of scientists and conspiracy belief, despite distrust being a strong individual-level predictor.

Suggested improvement: It would be beneficial to expand the discussion on why democracy moderates the effect of knowledge, but not of distrust, especially since mistrust is such a central predictor. This could expand the theoretical contribution of the manuscript.

2- Nuance in Attitudinal Variables

The result that “blind faith in science” (BLINDFAITH) positively correlates with SCT belief is interesting. The authors interpret this as a difference between trust in scientific methods and distrust in those who carry them out (the “academic elite”).

Suggested improvement: It would be beneficial to elaborate further on how “trust” or “faith” in science can be more precisely measured, or how naive faith manifests behaviorally among conspiracy believers. Exploring these facets in future research could deepen the field’s understanding.

3- Lower Explanatory Power of the “Science-in-Society” Model

The “Science-in-Society” model, which emphasizes engagement, accounted for less intra-country variance (0.8%) compared to the “Public Understanding” (10.2%) and “Science Literacy” (5.8%) models.

Suggested improvement: While not a weakness of the study itself (as it accurately reports the findings), It would be beneficial to offer a brief discussion on why this model, which often emphasizes public participation, might have less direct explanatory power for conspiracy belief in this context, or under what conditions its influence might be more substantial.

4- Interpretation of Democracy versus Other Country-Level Predictors:

The main result is that democracy at the country level is negatively associated with belief in science-related conspiracy theories. However, democracy is highly correlated with other macro-level factors such as GDP, corruption, and government effectiveness. This collinearity means that the effect attributed to democracy may not be truly independent. The authors mention this, but it would be beneficial to make this caution even more explicit, especially in the interpretation and implications sections. The authors may consider noting more clearly that the current models cannot fully separate the effect of democracy from these other macro factors.

5- Omission of Education as a Control

Education is a well-established predictor of conspiracy beliefs in prior research. In this paper, the authors use scientific knowledge ("Scientific Literacy") as their primary explanatory variable, rather than including general education as a control. This is a plausible decision since scientific literacy is closely related to formal education and directly aligns with the study’s theoretical framework. The authors position "Scientific Literacy" as a more targeted measure of the knowledge component under investigation.

However, since education is so commonly used as a control in literature, its omission limits the comparability of these results with previous studies. The authors may consider briefly clarifying their decision not to include education. A short discussion of this point may increase transparency and help readers situate the findings within the broader research field.

In conclusion, this article provides significant insights into the relationship between scientific literacy, trust in science, and the role of democracy in shaping belief in science-related conspiracy theories. While the paper is already strong in its design and execution, some targeted clarifications will make the contribution more transparent and accessible to a broad audience.

Reviewer #2: At present, the manuscript reads as though it contains two separate papers. The first is centered on individual-level variables, supported by its own introduction, literature review, and theoretical framework. The second brings in country-level variables, but lacks the necessary conceptual scaffolding—there is no clear introduction, literature review, or theoretical grounding for how country-level factors, particularly the quality of democracy, interact with individual-level science-related attitudes and conspiracy theory beliefs.

While the idea of integrating both levels of analysis through a multilevel modeling (MLM) approach is promising and could add value to the field, the absence of a unified theoretical framework tying together democracy, conspiracy thinking, and science skepticism represents a significant limitation. Given this conceptual gap, I recommend a major revise and resubmit (R&R) to restructure the manuscript and build a more coherent theoretical bridge between the individual and national levels of analysis.

The move from individual-level to country-level analysis feels like a shift between two distinct papers. While the effort to bridge levels of analysis is commendable, the integration is currently weak. I suggest either (1) strengthening the conceptual connection between individual and national variables or (2) focusing on one level of analysis to improve clarity and coherence.

Importantly, the manuscript needs to include the exact wording of the items used to measure both independent and dependent variables. Without this, it is difficult to evaluate construct validity.

The second half of the paper—including the modeling and results—is strong and well-executed. However, the first half of the manuscript requires substantial revision to improve clarity, flow, and theoretical coherence. The main contribution appears to lie in the interaction between individual-level attitudes and country-level democratic quality. Yet democracy is barely mentioned until later in the manuscript. I recommend reorienting the paper around this theme and condensing less relevant material.

1. Theoretical Framing and Literature Review

The theoretical positioning of the manuscript would benefit from greater cohesion. Specifically, the transition from individual-level variables (e.g., mistrust in science, knowledge levels) to country-level variables (e.g., quality of democracy) is not adequately signaled in the introduction. Readers are left uncertain as to how these levels interact, or whether they are intended to form an integrated theoretical model.

Moreover, the literature review (lines 81–161) is overly long and diffuse. I recommend condensing this section into a few focused paragraphs, concentrating specifically on prior work addressing science skepticism and science-related conspiracy theories. Excessive elaboration on epistemic authority and general background material could be streamlined to strengthen the paper’s contribution.

On a related note, the authors briefly mention “science-related populism” (line 60), but should clarify that not all populist movements are anti-science. Some populist actors have embraced scientific discourse when it aligns with their agenda. See Bayar and Seyis (2023) for further elaboration on context-dependent populist approaches to science-related policy making.

2. Claims, Citations, and Conceptual Clarity

In the abstract (lines 25–29), the authors claim that low levels of knowledge are correlated with belief in science-related conspiracy theories. However, this finding appears to contradict prior work such as Miller et al. (2015, AJPS), which shows that individuals with higher knowledge but low trust are more prone to conspiracy thinking. Are the authors referring specifically to science-related CTs here, or to a different subset of studies? Clarification is needed.

Additionally, the authors rely on numerical citations, which make it difficult to track specific claims or supporting literature. I strongly recommend the use of author-date in-text citations instead.

A sentence on lines 136–137—“this relationship decreases or even disappears when it comes to controversial scientific domains that have an impact on citizenship”—requires clarification. The authors should explain how this finding connects to their broader argument.

3. Hypotheses and Model Structure

The hypotheses (H1 and H2) appear to rely on variables that may be highly correlated (e.g., science skepticism and pro-science attitudes). If these variables share considerable conceptual overlap, the rationale for treating them as separate hypotheses should be better justified. Otherwise, they risk measuring the same latent construct.

The connection between the three models of STS and the authors' empirical approach is underdeveloped. It remains unclear how these models inform the analytical strategy or contribute to interpreting the results.

Finally, line 246 includes the phrase “prove,” and line 344 refers to findings as “true.” These should be revised to align with scientific norms—research supports or falsifies hypotheses; it does not prove them.

4. Data, Measurement, and Controls

The discussion of control variables (lines 256–279) could be reduced to a sentence or two. More importantly, the manuscript needs to include the exact wording of the items used to measure both independent and dependent variables. Without this, it is difficult to evaluate construct validity.

In discussing country-level analysis, the authors acknowledge the potential issue of collinearity among national-level variables (lines 550–551), but they do not explain how they addressed this statistically. How do they rule out competing explanations, such as economic development, GDP per capita, or mean education levels?

5. Terminology

The manuscript refers to conspiracy theory beliefs using inconsistent terminology. In line with common usage in the field, I recommend adopting the more standardized term “conspiracy theory beliefs” throughout, rather than alternatives like “beliefs in conspiracies” or “conspiracy beliefs,” for consistency and clarity.

The manuscript also uses multiple abbreviations (e.g., SCT, STS, TC, SSST) without consistently defining or reinforcing their meaning. For clarity and reader orientation, I strongly suggest the authors standardize their abbreviations and reintroduce them as necessary.

Additionally, although the discussion mentions “scientific populism,” this concept does not play a central role in the manuscript and is not sufficiently integrated into the core argument. The authors may want to either develop this thread further or remove it for focus.

Reviewer #3: Dear authors,

Thank you for giving me the opportunity to review your manuscript. I believe this is excellent research and only have a few comments.

Line 61: the first quotation mark does not have a closing quotation mark (")

Line 63: academic elite's (missing apostrophe)

Line 63: sovereignty (misspelled)

Line 246: the authors use the word "prove"; this is not a word that should be employed in empirical research as it is impossible to prove something with 100% certainty

Line 344: inconsistency in the numbering of hypotheses

Line 383: the equation is not clear (the parentheses are not appropriately sized)

Lines 452-455: the whole sentence is awkward

Line 457: The authors refer to "works" but do not cite any sources

Note for the hypotheses: the authors present the hypotheses with little to no text referring to these hypotheses in the main text. Usually when we include numbered hypotheses, we need to refer to them directly in the text. The authors do so for hypothesis 3 but not for the others.

Apart from these minor comments, I think this manuscript looks great and will make a good contribution to the research agenda on conspiracy beliefs, particularly through the multilevel and cross-national approach.

Do you want your identity to be public for this peer review? For information about this choice, including consent withdrawal, please see our Privacy Policy

Reviewer #1: No

Reviewer #2: No

Reviewer #3: No

---

## [Author Response · Author response to Decision Letter 1]

20 Oct 2025

Dear Dr. Erisen,

We would like to sincerely thank you for your constructive and thoughtful guidance, as well as the reviewers for their detailed and insightful feedback. We greatly appreciate the opportunity to revise and resubmit our manuscript “Belief in science-related conspiracy theories is not just a matter of knowledge: the democratic quality of countries as a protective factor.”

Following your recommendations and those of the reviewers, we have undertaken a thorough revision aimed at improving the conceptual coherence and theoretical integration of the paper. The revised version now articulates more clearly how individual-level factors—such as scientific knowledge, trust, and engagement—interact with contextual variables, particularly the democratic quality of countries. We have substantially rewritten the introduction and theoretical framework to make this multilevel logic explicit from the outset and to ensure a consistent connection between our analytical model and its theoretical underpinnings.

We have also strengthened the discussion and interpretation of our findings, expanding on the moderating role of democracy and providing a more nuanced explanation of how structural conditions influence the impact of knowledge and trust on science-related conspiracy theory beliefs. In addition, we have refined the presentation of our hypotheses, clarified the measurement of variables, and improved the consistency of terminology throughout the manuscript.

Overall, we believe that the revised manuscript now offers a much more coherent and theoretically grounded analysis of science-related conspiracy theory beliefs. We are very grateful for your and the reviewers’ insightful feedback, which has been invaluable in strengthening the quality and clarity of our work.

Kind regards,

Dr. Irene López-Navarro

Dr. Libia Santos-Requejo

------

Reviewer 1

Areas for Improvement

1- Partial Confirmation of Hypotheses

1.1 Hypothesis 3 (Engagement in Science): This hypothesis received only partial support. Engagement in scientific dissemination activities (ENGAGE) showed the expected negative correlation with SCT belief, but, interestingly, a favorable attitude towards citizen involvement in science decision-making (INVOLVEM) was linked to higher belief in SCTs.

Suggested improvement: Expanding the discussion around this unexpected result would be beneficial. The authors interpret this pattern as reflecting “alternative epistemologies” and dissatisfaction with academic elites. However, further theoretical exploration and/or qualitative research could clarify why greater support for lay involvement is associated with increased conspiracy belief. The authors may consider strengthening this part of the discussion by referencing the growing literature on science populism and anti-expert sentiment to clarify whether this association reflects a genuine demand for participatory democracy or, rather, a reaction against elites. It would be beneficial to cite recent studies and deepen the interpretation, enriching the discussion and situating the findings within broader debates.

In addition, recent scholarship shows a growing debate on the link between conspiracy beliefs and civic or political engagement. While some studies, such as Jolley and Douglas (2014) argue that conspiracy beliefs may suppress participation, others, like Kim (2022), find that conspiracy beliefs can stimulate engagement. In this paper, the unexpected positive association between support for citizen involvement in science decision-making and SCT belief (H3) could be better supported by directly referencing these contrasting perspectives in the literature. This would help readers understand such findings. However, at first counterintuitive, it aligns with emerging evidence that conspiracy beliefs sometimes motivate individuals to seek greater influence and involvement, often as a reaction to perceived exclusion by experts or institutions.

We appreciate this insightful comment. In the revised manuscript, we have substantially expanded the discussion of this result (see lines 440-477). Specifically, we now address the apparently paradoxical association between lower conspiracy belief among those engaged in science dissemination activities and higher support for citizen involvement among conspiracy believers. We situate this finding within the emerging literature on science-related populism and anti-expert sentiment (Mede & Schäfer, 2020; Merkley, 2020 and Mede et al., 2022), interpreting it as a tension between epistemic distrust and demands for participatory inclusion.

Furthermore, we incorporated references to the ongoing debate on the relationship between conspiracy beliefs and civic engagement (Jolley & Douglas, 2014 and Kim, 2022, as reviewer suggested, but also Imhoff et al., 2021; Halpern et al., 2019 and Bordeleau et al. 2023) to clarify that this positive association may reflect a reactive rather than a purely democratic form of engagement. These additions aim to strengthen the theoretical grounding of our interpretation and better align it with the reviewer’s valuable suggestions.

1.2 Hypothesis 5 (Moderation Effect): This hypothesis was only partially supported. The moderating effect of democracy was apparent in the relationship between scientific knowledge and SCT belief, but not in the link between distrust of scientists and conspiracy belief, despite distrust being a strong individual-level predictor.

Suggested improvement: It would be beneficial to expand the discussion on why democracy moderates the effect of knowledge, but not of distrust, especially since mistrust is such a central predictor. This could expand the theoretical contribution of the manuscript.

We appreciate this insightful suggestion. In the revised manuscript, we have expanded the discussion of Hypothesis 5 (lines 526-539) to provide a more detailed theoretical interpretation of the asymmetric moderating effect observed. Specifically, we now argue that the moderating role of democracy appears only for scientific knowledge because this variable depends on contextual conditions—such as educational systems, transparency, and access to reliable information—that are typically stronger in consolidated democracies. Therefore, democratic quality can enhance the protective role of knowledge against conspiracy thinking by fostering environments where citizens are better equipped to evaluate evidence and resist misinformation.

By contrast, distrust of scientists seems to operate primarily as an affective or identity-based orientation, less sensitive to institutional conditions and more strongly shaped by cultural, ideological, or populist narratives. As a result, such distrust may persist even in democratic contexts, where pluralism and freedom of expression can allow anti-elitist discourses to flourish. This reasoning clarifies why democracy moderates the effect of knowledge but not that of distrust.

2- Nuance in Attitudinal Variables

The result that “blind faith in science” (BLINDFAITH) positively correlates with SCT belief is interesting. The authors interpret this as a difference between trust in scientific methods and distrust in those who carry them out (the “academic elite”).

Suggested improvement: It would be beneficial to elaborate further on how “trust” or “faith” in science can be more precisely measured, or how naive faith manifests behaviorally among conspiracy believers. Exploring these facets in future research could deepen the field’s understanding.

We thank the reviewer for this insightful suggestion. We agree that the construct of “blind faith in science” deserves further conceptual and empirical refinement. In the present study, the variable BLINDFAITH captures a generalized, uncritical confidence in science’s problem-solving capacity, without necessarily distinguishing between trust in scientific methods and trust in scientific institutions or actors. Future research could extend this by incorporating behavioral or attitudinal indicators of naive scientism, such as unconditional acceptance of scientific claims, overestimation of science’s capacity to resolve moral or social issues, or resistance to legitimate critique of scientific practice. We have added a brief note on this in the Discussion section to acknowledge this limitation and potential avenue for future inquiry (lines 562-567).

3- Lower Explanatory Power of the “Science-in-Society” Model

The “Science-in-Society” model, which emphasizes engagement, accounted for less intra-country variance (0.8%) compared to the “Public Understanding” (10.2%) and “Science Literacy” (5.8%) models.

Suggested improvement: While not a weakness of the study itself (as it accurately reports the findings), It would be beneficial to offer a brief discussion on why this model, which often emphasizes public participation, might have less direct explanatory power for conspiracy belief in this context, or under what conditions its influence might be more substantial.

We thank the reviewer for this valuable suggestion. We agree that the relatively low explanatory power of the “Science-in-Society” model deserves additional reflection. As we now note in the Discussion section, one possible explanation is that attitudes related to participation and the democratisation of science (as captured by the Science-in-Society model) reflect normative ideals about how science should interact with society, rather than direct beliefs about how science actually functions. As a result, these attitudes may have a more indirect or contextual influence on conspiracy beliefs, depending on the level of institutional trust and the perceived openness of scientific governance in each country. We have added a short paragraph addressing this point in the Discussion (see lines 440-447).

4- Interpretation of Democracy versus Other Country-Level Predictors:

The main result is that democracy at the country level is negatively associated with belief in science-related conspiracy theories. However, democracy is highly correlated with other macro-level factors such as GDP, corruption, and government effectiveness. This collinearity means that the effect attributed to democracy may not be truly independent. The authors mention this, but it would be beneficial to make this caution even more explicit, especially in the interpretation and implications sections. The authors may consider noting more clearly that the current models cannot fully separate the effect of democracy from these other macro factors.

We appreciate the reviewer’s thoughtful observation. We agree that the high intercorrelation between democracy and other structural indicators (such as GDP per capita, corruption, and government effectiveness) limits our ability to isolate the independent effect of democracy. We have now made this limitation more explicit in both the Discussion section (lines 496-500 and 553-558). The revised text acknowledges that, although democracy shows a significant negative association with science-related conspiracy beliefs, this relationship may also reflect the broader institutional, economic, and governance environment typically associated with democratic societies.

5- Omission of Education as a Control

Education is a well-established predictor of conspiracy beliefs in prior research. In this paper, the authors use scientific knowledge ("Scientific Literacy") as their primary explanatory variable, rather than including general education as a control. This is a plausible decision since scientific literacy is closely related to formal education and directly aligns with the study’s theoretical framework. The authors position "Scientific Literacy" as a more targeted measure of the knowledge component under investigation.

However, since education is so commonly used as a control in literature, its omission limits the comparability of these results with previous studies. The authors may consider briefly clarifying their decision not to include education. A short discussion of this point may increase transparency and help readers situate the findings within the broader research field.

We appreciate this observation and fully agree that educational level has traditionally been used as a control variable in the literature on conspiracy beliefs. In our study, we decided not to include it for two main reasons:

First, due to issues of data quality and cross-national comparability. The Eurobarometer 516 includes two education-related items: the age at which respondents completed full-time education and the highest level of education attained, the latter recodable to ISCED levels. However, in our dataset, the variable on age at completion showed implausible values (up to 90 years or more), which would have required arbitrary recoding and data cleaning decisions to ensure reliability. Moreover, the heterogeneity of national education systems limits comparability even when approximating ISCED levels, as equivalence margins are not always consistent across countries.

Second, for theoretical and statistical reasons, educational level is strongly correlated with both scientific literacy and attitudes toward science and technology, which are the core constructs of our theoretical framework. Including education as a control could have introduced collinearity and redundancy, obscuring the specific contribution of scientific knowledge and attitudes. This relationship between formal education, scientific knowledge, and pro-science attitudes has been widely documented (Sturgis & Allum, 2004; Allum et al., 2008; Bak, 2001; Evans & Durant, 1995; Bauer et al., 2007).

To enhance transparency, we have added an explanatory note in the “Individual control variables” section (lines 229-235).

Reviewer 2

1. Theoretical Framing and Literature Review

The theoretical positioning of the manuscript would benefit from greater cohesion. Specifically, the transition from individual-level variables (e.g., mistrust in science, knowledge levels) to country-level variables (e.g., quality of democracy) is not adequately signaled in the introduction. Readers are left uncertain as to how these levels interact, or whether they are intended to form an integrated theoretical model.

We thank the reviewer for this valuable and insightful comment. We fully agree that the original version did not clearly articulate how individual- and country-level factors are theoretically integrated. In response, we have revised the manuscript to make the multilevel theoretical structure more explicit and coherent across sections.

First, we have added a new bridging paragraph at the end of the Introduction (lines 75–101) that explicitly connects individual predispositions with contextual environments. This paragraph highlights that attitudes toward science are shaped not only by knowledge and trust but also by the institutional and political settings in which they develop, thereby signaling the shift from micro- to macro-level analysis.

Second, in the theoretical framework (lines 165-170 and 206-210), we have expanded the section introducing country-level variables. We now explain in greater detail how political environments, and particularly the quality of democracy, provide the informational and normative context that shapes the effects of individual factors such as trust and scientific knowledge. This revision clarifies the rationale for testing democracy as a moderating variable in our model.

Finally, we have added a new integrative paragraph in the Discussion (lines 540-547) to synthesize how individual and contextual determinants jointly explain belief in science-related conspiracy theories. This paragraph emphasizes that democratic quality strengthens the protective influence of individual-level factors, thereby offering a more cohesive and theoretically grounded interpretation of our findings.

We believe that these revisions substantially improve the manuscript’s conceptual integration and address the reviewer’s concern regarding the coherence and clarity of the theoretical framework.

Furthermore, one of the novel contributions of our analysis is precisely the examination of conspiracy theory beliefs from a multilevel perspective, incorporating structural variables that have traditionally remained outside the scope of most explanatory models. A

---

## [Decision Letter · Decision Letter 1]

24 Nov 2025

Belief in science-related conspiracy theories is not just a matter of knowledge: the democratic quality of countries as a protective factor

PONE-D-25-29874R1

Dear Dr. López-Navarro,

We’re pleased to inform you that your manuscript has been judged scientifically suitable for publication and will be formally accepted for publication once it meets all outstanding technical requirements.

Kind regards,

Cengiz Erisen

Academic Editor

PLOS ONE

Additional Editor Comments (optional):

Reviewers' comments:

Reviewer's Responses to Questions

**Comments to the Author**

Reviewer #1: All comments have been addressed

Reviewer #3: All comments have been addressed

2. Is the manuscript technically sound, and do the data support the conclusions?

Reviewer #1: Yes

Reviewer #3: Yes

3. Has the statistical analysis been performed appropriately and rigorously?

Reviewer #1: Yes

Reviewer #3: Yes

4. Have the authors made all data underlying the findings in their manuscript fully available?

Reviewer #1: Yes

Reviewer #3: Yes

5. Is the manuscript presented in an intelligible fashion and written in standard English?

Reviewer #1: Yes

Reviewer #3: Yes

Reviewer #1: I acknowledge the authors' response to the initial review. The revisions provided in the rebuttal letter and the corresponding edits in the text address the concerns raised. The manuscript has been improved, particularly in terms of the theoretical framework and the interpretation of the statistical findings. The inclusion of new literature regarding science populism and the articulation of the distinction between institutional and affective dimensions of trust have clarified the paper’s contribution.

Below, I provide a detailed assessment of how the specific points have been addressed:

1. Hypothesis 3 (Engagement in Science) In the initial review, I noted the paradoxical finding regarding the positive correlation between support for citizen involvement in science decision-making and belief in science-related conspiracy theories. The authors have expanded the discussion (lines 440–477) to address this issue.

• Theoretical Integration: The integration of the "science populism" framework (referencing Mede & Schäfer; Merkley) and the inclusion of the debate on civic engagement (comparing Jolley & Douglas vs. Kim) provides the necessary theoretical context.

• Interpretation: Interpreting this finding as a tension between "epistemic distrust" and "participatory demands" offers a logical resolution. The manuscript now positions this result as evidence of "alternative epistemologies" where conspiracy believers seek to reclaim power from perceived elites. This revision clarifies the findings within the study's framework.

2. Hypothesis 5 (The Moderating Role of Democracy) The authors were asked to clarify why democracy moderates the effect of scientific knowledge but not the effect of distrust. The expanded discussion (lines 526–539) offers a plausible explanation.

• Distinction of Mechanisms: The authors argue that scientific knowledge is "context-dependent," shaped by educational quality, transparency, and institutional reliability—conditions that are generally stronger in consolidated democracies. By contrast, distrust in scientists is characterized as an "affective or identity-based orientation," which is less shaped by the institutional environment and more by cultural or populist narratives.

• Assessment: This theoretical distinction is sound. It explains why "affective distrust" persists in democratic contexts, providing a consistent explanation for the null finding regarding the moderation of distrust.

3. Blind Faith in Science (BLINDFAITH) The authors have addressed the comments regarding the conceptual limitations of the "Blind Faith" variable.

• Assessment: They have added a clarification acknowledging that the current variable captures a generalized confidence rather than a specific trust in methodology (lines 557–567). The outline of behavioral indicators for "naïve scientism" is a useful addition to the discussion for future research.

4. Low Explanatory Power of the Science-in-Society Model The revised text (lines 440–447) provides a clear explanation for the model’s lower explanatory power (0.8%).

• Assessment: Clarifying that Science-in-Society variables capture "normative beliefs" (how science should function) rather than beliefs about actual scientific practice is a valid interpretation. Acknowledging that these variables may influence SCT belief only indirectly and depend on national context places the findings in the correct perspective.

5. Democracy vs. Other Country-Level Predictors The authors have updated the discussion in several places (lines 496–500, 553–558) regarding the collinearity between democracy and other macro-factors.

• Assessment: The explicit statement that democracy is highly correlated with GDP, corruption, transparency, and governance indicators addresses the methodological concern. By acknowledging that the current model cannot fully isolate democracy’s independent causal effect from broader institutional environments, the interpretation is methodologically transparent.

6. Omission of Education as a Control Variable I accept the explanation provided in the methodology section (lines 229–235) for omitting education as a control variable.

• Assessment: The justification based on data quality issues in the Eurobarometer education items (including implausible values) and the potential multicollinearity with scientific literacy is valid. The decision to focus on scientific literacy is appropriate given the study's theoretical focus and data constraints.

Conclusion: The authors have addressed the theoretical and methodological points raised during the review process. The revised manuscript presents a multi-level analysis that is grounded in the relevant literature.

Reviewer #3: Dear authors,

Thank you for responding to my comments. I am more than satisfied with the changes made and look forward to this research making an important contribution to the field.

**Do you want your identity to be public for this peer review?** For information about this choice, including consent withdrawal, please see our Privacy Policy

Reviewer #1: **Yes: ** Dilale Dönmez

Reviewer #3: No

---

## [Editor Report · Acceptance letter]

PONE-D-25-29874R1

PLOS One

Dear Dr. López-Navarro,

I'm pleased to inform you that your manuscript has been deemed suitable for publication in PLOS One. Congratulations! Your manuscript is now being handed over to our production team.

Kind regards,

on behalf of

Dr. Cengiz Erisen

Academic Editor

PLOS One